# A Wrinkled Ag/CNTs-PDMS Composite Film for a High-Performance Flexible Sensor and Its Applications in Human-Body Single Monitoring

**DOI:** 10.3390/nano9060850

**Published:** 2019-06-03

**Authors:** Yanpeng Yang, Chengzhi Luo, Junji Jia, Yafei Sun, Qiang Fu, Chunxu Pan

**Affiliations:** 1Shenzhen Research Institute, Wuhan University, Shenzhen 518057, China; yangshu@whu.edu.cn (Y.Y.); luo_cz@foxmail.com (C.L.); fuqiang@whu.edu.cn (Q.F.); 2School of Physics and Technology, and MOE Key Laboratory of Artificial Micro- and Nano-Structures, Wuhan University, Wuhan 430072, China; 3School of Electronic and Computer Engineering, Peking University, Shenzhen 518005, China; 4Center for Theoretical Physics, Wuhan University, Wuhan 430072, China; junjijia@whu.edu.cn; 5Mechanical and Electrical Engineering College, Shenzhen Institute of Information Technology, Shenzhen 518172, China; sunyf@sziit.edu.cn; 6Center for Electron Microscopy, Wuhan University, Wuhan 430072, China

**Keywords:** sensor, CNTs, Ag film, force-electric effect, human health monitoring

## Abstract

In this paper, a flexible Ag/CNTs-PDMS (polydimethylsi-loxane) composite film sensor based on the novel design philosophy was prepared. Its force-electric effect mechanism is based on the generation of micro-cracks in the Ag film during external forcing, leading to resistance variation. Experimental results find that Ag film thickness has a strong influence on the sensor’s sensitivity, which exhibits a tendency of first increasing and then decreasing the Ag film thickness, and also has an optimal thickness of 4.9 μm for the maximum sensitivity around 30. The sensitive mechanism can be theoretically explained by using the quantum tunneling effect. Due to the use of the wrinkled carbon nanotubes (CNTs) film, this sensor has advantages, such as high sensitivity, large strain range, good stability and durability, cheap price, and suitability for large-scale production. Preliminary applications on human-body monitoring reveal that the sensor can detect weak tremors and breathe depth and rate, and the corresponding heartbeat response. It provides possibilities to diagnose early Parkinson’s disease and exploit an early warning system for sudden infant death syndrome and sleep apnea in adults. In addition, as a force-electric effect sensor, it is expected to have broad application areas, such as a man-machine cooperation, and a robotic system.

## 1. Introduction

Currently, monitoring human physiological signals is regarded as an effective method for disease diagnosis and health assessment [1]. The wearable and flexible medical sensor has attracted much attention in the prospect of monitoring the general well-being of interested users, the real-time performance of athletes, or the disease status of patients [2,3]. The use of this device also encourages people to take a greater interest in their own healthcare in a more convenient and cheaper way and thus improves their compliance. In general, such devices monitor various kinds of bio-signals from human beings through perspiration, epidemic skin, breath, urine, and saliva [4,5,6,7,8]. It appears the design and development of wearable sensor systems for measuring and quantifying physical signals generated by the human body provide an opportunity for disease diagnosis, therapy, and health monitoring [9,10,11,12,13,14].

Up to now, great progress has been achieved in this area. Lee et al. [15] proposed a novel sensor for sweat-based diabetes monitoring and feedback therapy based on functional graphene. The integrated system consisted of the following modules: sweat-control components, sensing components and therapeutic components. The orchestrated monitoring of bio-markers and physiological cues with sweat control and transcutaneous drug delivery achieved a closed-loop, point-of-care treatment for diabetes. Yamamoto et al. [4] integrated an electrocardiogram sensor and a temperature sensor as a simple flexible sensor system to monitor health condition change according to the electrocardiogram signal, and dehydration and heat stroke via skin temperature variation. Schwartz et al. [16] prepared a flexible polymer transistor with high pressure sensitivity for application in electronic skin and health monitoring. They also demonstrated that this sensor could be used for non-invasive, high fidelity, continuous radial artery pulse wave monitoring, which might lead to the use of flexible pressure sensors in mobile health monitoring and remote diagnostics in cardiovascular medicine. Myung et al. [17] developed a graphene-encapsulated nanoparticle-based biosensor for the selective detection of cancer biomarkers. This biosensor could easily detect any vital cancer biomarkers and had a high selectivity and sensitivity. The ease of fabrication and biocompatibility, along with excellent electrochemical and electrical properties of the graphene nano-composites, made it an ideal candidate for future bio-sensing application in a clinical setting.

A sensor is a device that converts a physical signal into a measurable electrical signal. According to different structures and principles, the commonly used sensors include temperature sensors, pressure sensors, displacement sensors, strain sensors, chemical sensors, and biosensors. For a desired high-performance sensor, high sensitivity and long service life play two crucial rules. In recent decades, nanomaterials have provided new opportunities for sensors, in which carbon nanotubes (CNTs) exhibit a great potential due to their excellent mechanical, thermal, and electrical properties [18,19,20,21]. Compared to regular materials for preparing sensors, CNTs have some irreplaceable advantages: (1) Extremely high length-diameter ratio (up to 104) and large specific surface area (>1500 m^2^/g), which provide the possibility for loading variant sensitive materials and making different types of sensors. (2) High stability at room temperature with excellent mechanical properties, which greatly improves the service life of the sensor. (3) High strength substrates for producing flexible and wearable sensors, which greatly expand the application range of traditional sensors. 

In our previous work [22], by inspiring the crack-shaped structure of a spider’s slit organs near its leg joints, we prepared a novel multifunctional Au/CNTs-PDMS composite film sensor with high sensitivity and durability. The key process was, firstly, making a wrinkle structure of the CNTs film upon the flat polymer substrate to act as the conducting network and, secondly, physically ion sputtering a crack-shaped Au film as the sensitive transducer. The working principle of this sensor was that the strain sensitivity was obtained from the electrical resistance variation during the Au film deformation, which caused the opening/closure of the Au film micro-cracks and obtained the force-electric effect, while the CNTs conductive film with wrinkle structure provided high sensitivity to small strain and large strain ranges.

Compared with gold (Au), silver (Ag) has special characteristics, such as low volatility, good ductility, slightly higher electrical and thermal conductivities, a relatively large storage capacity, and a cheap price, which make it more suitable for large-scale production. There has been a lot of researche into the sensor’s use of Ag nanoparticles or films. Lee et al. [23] firstly developed highly-sensitive, transparent, and durable pressure sensors based on sea-urchin-shaped metal nanoparticles (Au, Ag). This device could detect minute movements of human muscles, such as finger bending and hand motion. Takei et al. [24] prepared highly sensitive electronic whiskers based on patterned carbon nanotube and silver nanoparticle composite films. Ag NP (silver nanoparticles) ink and CNTs paste were mixed with tunable component concentrations, and the composite mixture was then patterned onto a polydimethylsiloxane (PDMS) substrate of desired shape and geometry by either painting or printing. This electronic whisker could detect minute pressure changes and its properties remained almost unchanged for 1000 cycles. Zaretski et al. [25] introduced a flexible sensor by using a process, i.e., evaporating metal nanoparticles upon graphene sheets, and formed graphene-based metal nanosized islands. They used this sensor to detect human health, such as pulse and cardiomyocyte contraction, etc.

Firstly, we transferred the CNTs film as a conductive material to the polydimethylsiloxane (PDMS) substrate and made the CNTs film into a wrinkle structure. Secondly, we sputtered a Ag film as a sensitive transducer. Lastly, we assembled a flexible Ag/CNTs-PDMS composite film sensor. The principle was that when the Ag/CNTs-PDMS composite film was stretched by an external force, the micro-cracks would be induced in the Ag film and led to resistance variation, which would produce a force-electric effect. In comparison to other sensors, the special CNTs wrinkle structure ensured the sensor high-sensitivity, large strain range, and long service life. To obtain the highest performance, the effects of different Ag film thicknesses on the sensitivity of the sensor was studied, and it was found that, according to the Ag film thickness increase, the sensitivity enhanced and then declined, i.e., there was an optimum value. In addition, by using this sensor, we continued preliminary monitoring of the human body signals, including subtle human motions, heartbeat, and breathing, which exhibited wide potential applications in the prophylactic medicine field, such as early diagnosis of Parkinson’s disease, monitoring and prevention of sudden infant death syndrome and sleep apnea in adults.

## 2. Materials and Methods

Figure 1 schematically illustrates the key steps for fabricating the flexible Ag/CNTs-PDMS composite film sensor. (1) We prepared CNTs in a self-made floating catalyst chemical vapor deposition (FC-CVD) system and the CNTs film was deposited on Al foil, and its thickness was about 100 nm. (2) We removed the CNTs film from the Al foil by etching in a 1 M HCl aqueous solution, and after the Al foil was dissolved, the CNT film floated on the HCl solution. (3) A PDMS substrate, with around 1 mm thickness, was fixed on a glass slide at one end, and a force was applied to the other end to stretch it for obtaining the pre-strained PDMS. (4) We transferred the CNTs film onto the pre-strained PDMS film and then released the pre-strained PDMS, which resulted in the CNT film becoming a wrinkle structure. (5) We physically deposited the Ag film upon the wrinkled CNTs film via ion sputtering (SBC-12, KYKY, Beijing, China), with the parameters involving distance 5 cm between the sample and the Ag target, and with a sputtering current of 6 mA. (6) We assembled the flexible and wearable sensor.

The surface and cross section morphologies of the Ag/CNTs-PDMS composite film were observed by using a scanning electron microscope (SEM, S-4800, Hitachi, Tokyo, Japan). The properties of the sensor were measured by an electrochemical workstation (CHI 660D, Shanghai Chenhua, Shanghai, China).

In order to monitor weak signals of human-body, the sensor was attached to different positions of the human body by using medical tapes, such as the face, throat, finger joints, upper lip, wrist, and heartbeat. Preliminarily, the sensor was attached to finger joints and the wrist to monitor the minute movements, with an interval time of 15 s and to the upper lip and heartbeat to monitor the conditions of normal breathing and simulated breathing-hold, with an interval time of 20 s.

## 3. Experimental Results and Discussion

Figure 2 shows the SEM morphologies of the original CNTs and the Ag/CNTs-PDMS composite films under variant tensile strain conditions. It could be seen that the original CNTs film was of a continuous network with a wrinkle structure before it was stretched and became flat when the tensile strain of 10% was applied, and vice versa during releasing, as shown in Figure 2a–e. In addition, it was worth paying attention to the fact that the Ag film deposited on the wrinkled CNTs showed a flat surface when no tensile strain was applied and many micro-cracks and islands appeared alongside the increasing of the tensile strain, and vice versa during releasing, as shown in Figure 2f–j.

It is well known that the change of the micro-cracks on the Ag film will inevitably generate the variation of its resistance, that is to say, the variation of electric current passing through the Ag film, which is the basic principle of the force-electric effect. However, in the present work, the Ag film was deposited on the specially-treated CNTs film surface with a wrinkled structure. Therefore, it would be very sensitive to any tiny alteration of a force action, i.e., once a very small stress-strain was applied to the Ag/CNTs-PDMS film, a large resistance or current response would be generated, which caused a high sensitivity [22]. In addition, from Figure 3 a local high magnification, the CNTs film kept an integrated structure beneath the Ag film crack. This was because the high strength and network of the CNTs film could inhibit the crack expansion and prevent it from completely cracking, which guaranteed the structural integrity of the overall structure of the sensor during stretching.

In fact, there are many factors to influence the sensitivity of a sensor. Besides the selection of materials and structure design, the thickness of the sensitive materials, such as the Ag film, is also a crucial factor, which, however, has not yet been reported.

Figure 4a–f shows the cross-sectional morphologies of the Ag film thicknesses upon the Ag/CNTs-PDMS composite films with the ion sputtering times at 0.5 min, 1 min, 2 min, 4 min, 6 min, and 8 min, respectively. The Ag film thickness increased with extension of the ion sputtering time. Figure 5a,b and Table 1 give the relationship between the Ag film thickness, the sputtering time, and the related original resistances. The Ag film thickness approximately exhibited a proportional liner relation to the ion sputtering time, which met the ion sputtering mechanism. In addition, in the case of the constant length and width of the films, we measured the values of original resistance R_0_, which were 3.6, 2.5, 1.75, 1.05, 0.60, and 0.55 kΩ, respectively. The relationship between the original resistance R_0_ and the Ag film thickness was also consistent with the formula $R=\frac{\rho L}{∆\mathrm{tw}}$ (where R is resistance, ρ is resistivity, L is length and w and t are the width and thickness of the Ag film, respectively. When the ρ, L, and w were unchanged, R was inversely proportional to t, which meant that R is proportional to 1/∆t). In the case of the short sputtering time (0.5 min), the large deviation was due to the over-thin Ag film.

In general, the metal film thickness holds influence over the sensitivity of the sensor. For example, Lee et al. [26] found that the performance of the crack sensor had a great relationship with the metal (Au, Cr) thickness. In order to study the effect of the Ag film thickness on the strain sensing performance of the Ag/CNTs-PDMS films, we measured the relationship between the relative resistance, △R/R_0_ = (R−R_0_)/R_0_ (where R_0_ and R are the resistance before and after the stretch, respectively), and the strain under the constant load voltage. Figure 6a–f illustrates the relative resistance variations (△R/R_0_) and gauge factors (GF) versus the strains from 0% to 50%, when the Ag film thicknesses were 0.5 μm, 1.5 μm, 1.75 μm, 3.26 μm, 4.9 μm, and 6.4 μm, respectively. For all Ag film thicknesses of the samples, △R/R_0_ increased, at first at a faster pace which then tended to a flat gradient until it approached the asymptotic value at the largest strain. These changes could be described by the quantum tunneling effect of the Ag film under strain [22]. As shown in Figure 6g, when the pure Ag film was without the CNTs support, the △R/R_0_ was too large to have a closed circuit, due to the cracks and fractures caused by 3% tensile strain, which led to an irreversible degradation of the Ag film conductance. This result also demonstrated the importance of the wrinkled CNTs film as a conduct network with excellent mechanical properties and high durability to the sensor.

Here, we derived the relationship between the total resistance and strain by using the same model:(1)$$R\left(\varepsilon \right)=\frac{1}{\frac{1}{R_{0}}+\frac{1}{R_{\mathrm{Ag}0}+R_{\mathrm{Agc}}(e^{c\varepsilon }-1)}}$$
where R_0_ is determined by the resistance of CNTs and the contact resistance between CNTs and the Ag film, R_Ag0_ is the resistance of the Ag film when there is no strain, R_Agc_ is the resistance factor of the quantum tunneling section, c is a factor related to the quantum tunneling effect, and ε is the strain. R_Ag0_ is inversely proportional to the total thickness of the Ag film, and R_Agc_ is proportional to the thickness of the Ag film torn by strain. By adopting Formula (1) and fitting $\frac{∆R}{R_{0}}\left(\varepsilon \right)$ in each sample, the values of R_0_, R_Ag0_, R_Agc_, and c were obtained, respectively. Therefore, the sensitivity of the sensor could be calculated according to the definition dRRdLL, as shown the red lines in Figure 6a–f. In other words, for each sensor, the sensitivity decreased gradually from the maximum value (when the strain was 0) to 0 with increase of strain, which also proved that the present sensor was more suitable for measuring tiny strains, i.e., for monitoring small strains in the human health status [27,28,29,30].

In order to study the effect of the Ag film thickness on sensor sensitivity (GF (ε = 0)), we take the example of a small strain. From Formula (1), we could get the following formula:(2)$$\mathrm{GF}\left(\varepsilon =0\right)=\frac{{c\;R}_{\mathrm{Agc}}}{R_{\mathrm{Ag}0}+R_{\mathrm{Ag}0}^{2}/R_{0}}$$

For each Ag film thickness of the sample, R_Ag0_ and R_Agc_ could be obtained by fitting R(ε), as shown in Figure 7. Figure 7a,b give the changes of 1/R_Ag0_ and 1/R_Agc_ with the Ag film thicknesses, respectively. It could be seen that the values of 1/R_Ag0_ and 1/R_Agc_ exhibited a linear relationship with the Ag film thicknesses (Δt), which also met the Ohm’s law of resistance. However, in the case of longer spurting time for a thicker Ag film thickness, there was a big deviation of 1/R_Agc_ from the linear relation, which might reflect the too simple quantum tunneling tandem model to explain the sample. Regarding the values of R_Ag0_ and R_Agc_ in Figure 7a,b, the relationship between the sensitivity GF (ε = 0) and the Ag film thickness could be obtained, as shown in Figure 7c. When the strain was 0, the sensitivity GF (ε = 0) emerged a tendency of first increasing and then decreasing with the Ag film thickness. Except for the last sample, there was a linear relationship between the sensitivity and thickness, which was consistent with the previous deduction [22]. Importantly, it was found that there existed an optimal Ag film thickness of 4.9 μm for the maximum sensitivity, which was around 30, which also demonstrated the great impact of the Ag film thickness on the sensor sensitivity. Compared to other sensors, the present Ag/CNTs-PDMS composite sensor achieved sensitivity up to 30 and withstood strain up to 50%, which makes it more suitable for application as a wearable strain sensor in terms of cost, sensitivity, and preparation process. For example, Yamada et al. [27] prepared a stretchable carbon nanotube strain sensor for human-motion detection, which exhibited advantages, such as strain toleration up to 280%, high durability, fast response, and low creep, but the sensitivity (GF = 0.8) was low. Kang et al. [3] proposed an ultrasensitive mechanical crack-based sensor inspired by the spider sensory system with sensitivity up to 2000, but it tolerated strain only up to 2%. In our previous work about an Au/CNTs-PDMS composite film sensor [22], sensitivity was up to 70, but the Au transducer layer employed was more expensive than the present Ag layer. In order to analyze the singular behavior of the sensitivity and R_Ag0_ value in the case of the thicker Ag film with a long sputtering time (8 min), we observed the crack morphologies under small strain, with sputtering times of 6 min for comparison, as shown in Figure 7d,e. It was found that the Ag film was completely cracked under a certain strain condition and could be seen that the bottom at the sputtering time was 6 min, which could confirm the occurrence of the quantum tunneling effect. However, the thick Ag film of the 8 min sputtering time showed an incomplete crack with a connected bottom, which resulted in the failure of the original quantum tunnel tandem resistance model and the R_Ag0_ unveracious data. In addition, the connected bottom led to the fact that a large portion of the Ag film thickness did not induce the quantum tunneling effect, which actually reduced the effective thickness and decreased the sensitivity. Experiments showed that the maximum tensile strain of the sensor was up to 45%, which greatly exceeded the regular 5% strain limit of the traditional metal or semiconductor sensors [31,32,33].

Durability and stability are also of crucial importance to a sensor. Cui et al. [34] reported a flexible pressure sensor with Ag wrinkled electrodes based on a PDMS substrate. The Ag wrinkled electrodes were formed by vacuum deposition on top of the pre-strained and relaxed PDMS substrates, which were treated using an O_2_ plasma, a surface functionalization process, and a magnetron sputtering process. The adhesion of the Ag to PDMS was improved by pretreatment of the polymer, and the mechanical stability was also provided. Regarding the present work, the measurements for the samples at different strains and Ag film thicknesses are shown in Figure 8, which illustrates the relative resistance variations under step strain from 0% to 17.4% strain, when the Ag film thicknesses were 0.5 μm, 1.5 μm, 1.75 μm, 3.26 μm, 4.9 μm, and 6.4 μm, respectively. It could be seen that when a step strain of 4.25% was applied, the relative resistance (△R/R_0_) of the composite films was kept almost unchanged in the platform, but sharply magnified at the step edge, and with the increase of the Ag film thickness, it showed the tendency to increase first and then decrease. In addition, Figure 9 illustrates the multicycle tests of the relative resistance (△R/R_0_) variations, when the Ag film thicknesses were 0.5 μm, 1.5 μm, 1.75 μm, 3.26 μm, 4.9 μm, and 6.4 μm, respectively. When 8.5% periodic strain was applied to each sensor, the △R/R_0_ curve variations were also periodic and the peak patterns remained no different with the same tendency regarding to different Ag film thickness. These results demonstrated the high stability of the Ag/CNTs-PDMS composite film sensor under the different Ag film thicknesses, due to the fact that the output signal was highly reproducible.

In order to evaluate the mechanical durability, Figure 10 shows the measurements of multiple strain cycles of the Ag/CNTs-PDMS composite film sensor under the condition of the Ag film thickness 4.9 μm, corresponding to the highest sensitivity at 0.05 Hz frequency. Figure 10a,b illustrates the relative resistance variations at 8.5% strain as a function of strain cycle, in which Figure 10a gives the relative resistance variation characteristic after 20 cycles and Figure 10b gives the relative resistance variation characteristic after 2000 cycles. Figure 10c reveals that, up to 5000 strain cycles, the ΔR/R_0_ values didn’t not have much fluctuation and their electrical performance had no significant decline, which showed that the sensor has excellent mechanical durability. As an important application for the flexible Ag/CNTs-PDMS composite film sensor, we tried to use it for monitoring the weak signal from the human body. It was attached to the various parts of the human body, such as fingers, wrists, chest, and upper lip, by using medical tapes, as shown in Figure 11. For example, the sensor was fixed onto the finger to monitor its response to tremors in different bending statuses, as shown in Figure 11b, where the red line and blue line represents the normal bending and the simulated tremor of the finger, respectively. Through comparison, it could be seen that if the finger was in tremor, the ΔR/R_0_ value would have a very apparent difference under certain step strains. Similarly, if the sensor was put onto the wrist, it was found that the simulated tremor had an apparent peak compared with the normal status, as shown in Figure 11c. In fact, a person who has a certain disease, such as early Parkinson’s disease, may have involuntary body tremors and sometimes the tremor may be too weak to be felt or observed by the sensor. The present flexible sensor with highly-sensitive and large strain range provides a possibility to diagnose and prevent these kinds of early diseases. This is our further ongoing research work.

In addition, when the sensor was attached to the upper lip, the human respiration process (exhaled gas flow) was recorded, in which the peak amplitude and frequency represented the breath depth and rate, respectively, and the ΔR/R_0_ variation was the rate of exhalation. It was found that, for normal breathing, there were larger peak differences between the highest peak and the lowest peak, while the peak differences became less when the holding-breathe was performed, as shown in Figure 11d. It was worth noting that there was also a great difference in peak patterns of heartbeat between the holding-breath and normal breathing when it was attached to the chest, as shown in Figure 11e. These experimental results told us that the sensor’s response to breathing and heartbeats could be used for preventing and monitoring some diseases. For example, regarding its high sensitivity, it could be attached to the upper lip or heartbeat and exploit an early warning system for sudden infant death syndrome and sleep apnea in adults.

## 4. Conclusions

A novel flexible Ag/CNTs-PDMS composite film sensor was prepared via the main process, i.e., firstly, transferring a CNTs film upon the stretched PDMS substrate and then releasing the substrate to obtain a wrinkle structure of the CNTs film; secondly, ion sputtering an Ag film with a certain thickness upon the CNTs film. The wrinkled CNTs film was used as the conducting network, and the Ag film as the sensitive transducer. Its force-electric effect mechanism was based upon the generation of micro-cracks in the Ag film during external forcing, leading to resistance variation.

The design philosophy of the wrinkled CNTs film gave the sensor high sensitivity and large strain range. It was found that the sensor’s sensitivity had a tendency of first increasing and then decreasing with the Ag film thickness, with optimal thickness of 4.9 μm for the maximum sensitivity around 30.

By using the CNTs film as a conducting network, the sensor’s stability and mechanical durability are greatly enhanced. The preliminary application on the human body single monitoring showed its potential in prophylactic medicine and as a warning system for some diseases. As a force-electric effect sensor, it is expected to have broad application areas, such as mechanical failure detection, web of Things, man-machine cooperation, and a robotic system.

## Figures and Tables

**Figure 1 nanomaterials-09-00850-f001:**
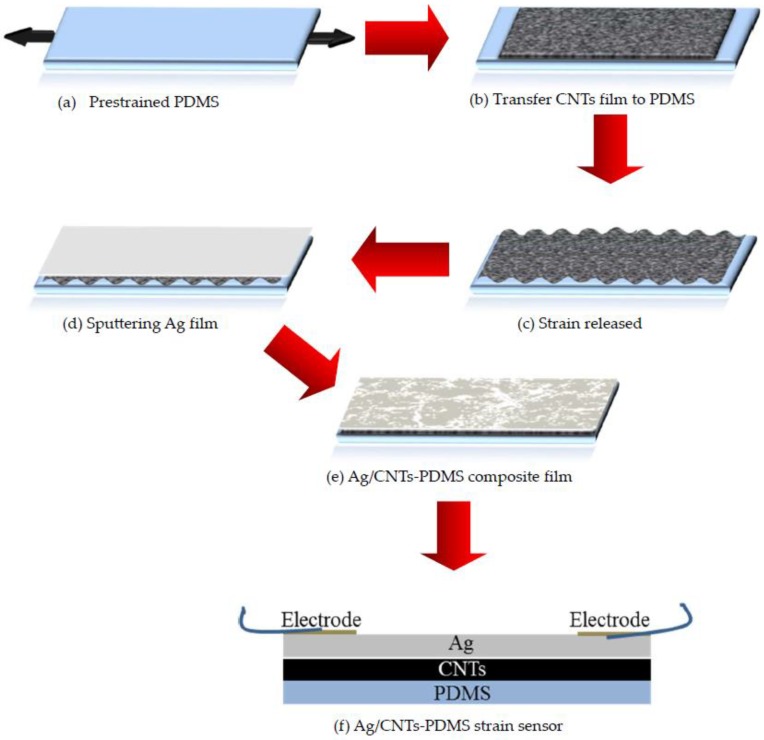
Schematic diagram of preparation steps for fabricating the Ag/CNTs-PDMS composite film sensor.

**Figure 2 nanomaterials-09-00850-f002:**
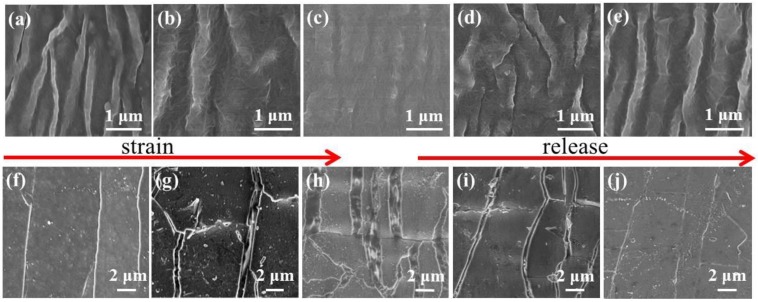
Scanning electron microscope (SEM) morphologies of the wrinkled original carbon nanotubes (CNTs) and the Ag/CNTs-PDMS composite films under variant strain and release conditions. (**a**–**e**) Original CNTs at tensile strain 0%, 5%, 10%, 5%, and 0%, respectively; (**f**–**j**) Ag/CNTs-PDMS composite films at tensile strain 0%, 5%, 10%, 5%, and 0%, respectively.

**Figure 3 nanomaterials-09-00850-f003:**
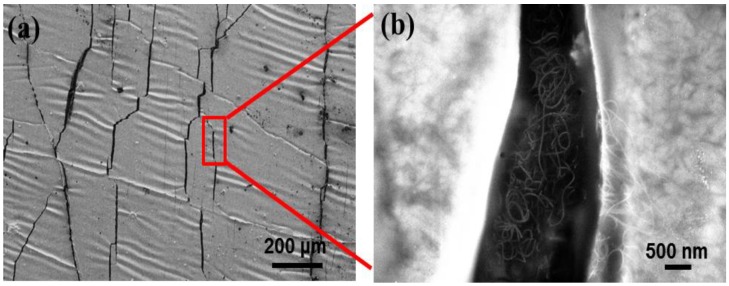
SEM morphologies of the Ag/CNTs-PDMS composite film surface. (**a**) Vertical view. (**b**) Local high magnifications image.

**Figure 4 nanomaterials-09-00850-f004:**
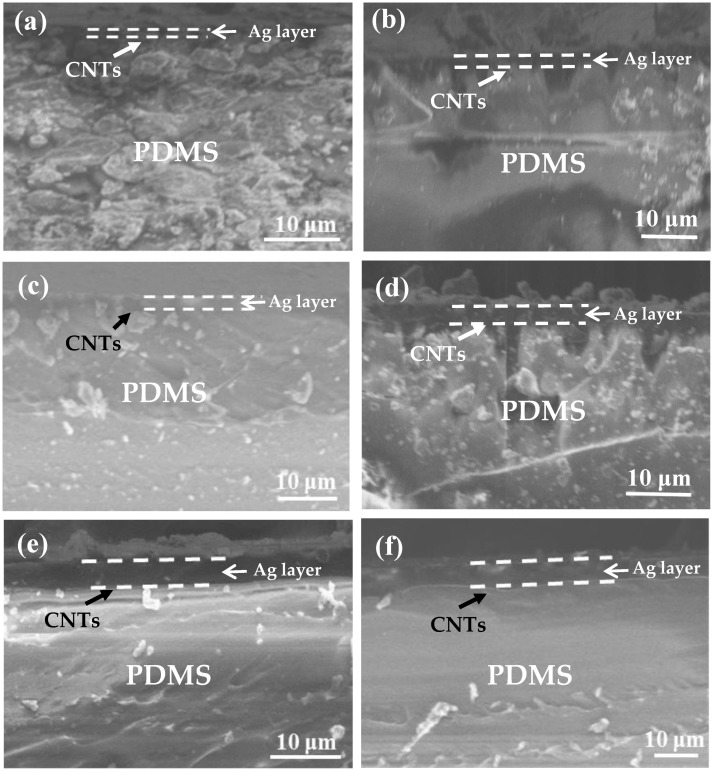
Thickness variations of the Ag film upon the Ag/CNTs-PDMS composite films at different ion sputtering times. (**a**) 0.5 min; (**b**) 1 min; (**c**) 2 min; (**d**) 4 min; (**e**) 6 min; (**f**) 8 min.

**Figure 5 nanomaterials-09-00850-f005:**
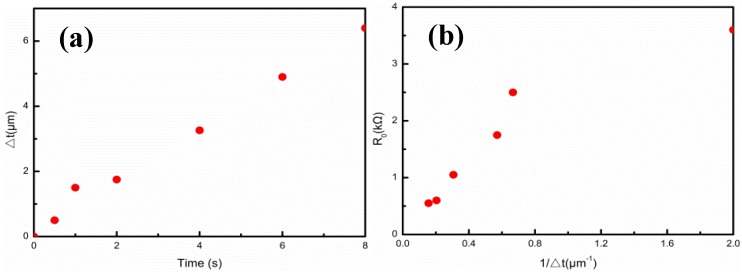
Relationships of the Ag film thicknesses (∆t) with sputtering times and original resistances (R_0_). (**a**) ∆t-Time; (**b**) R_0_-1⁄∆t.

**Figure 6 nanomaterials-09-00850-f006:**
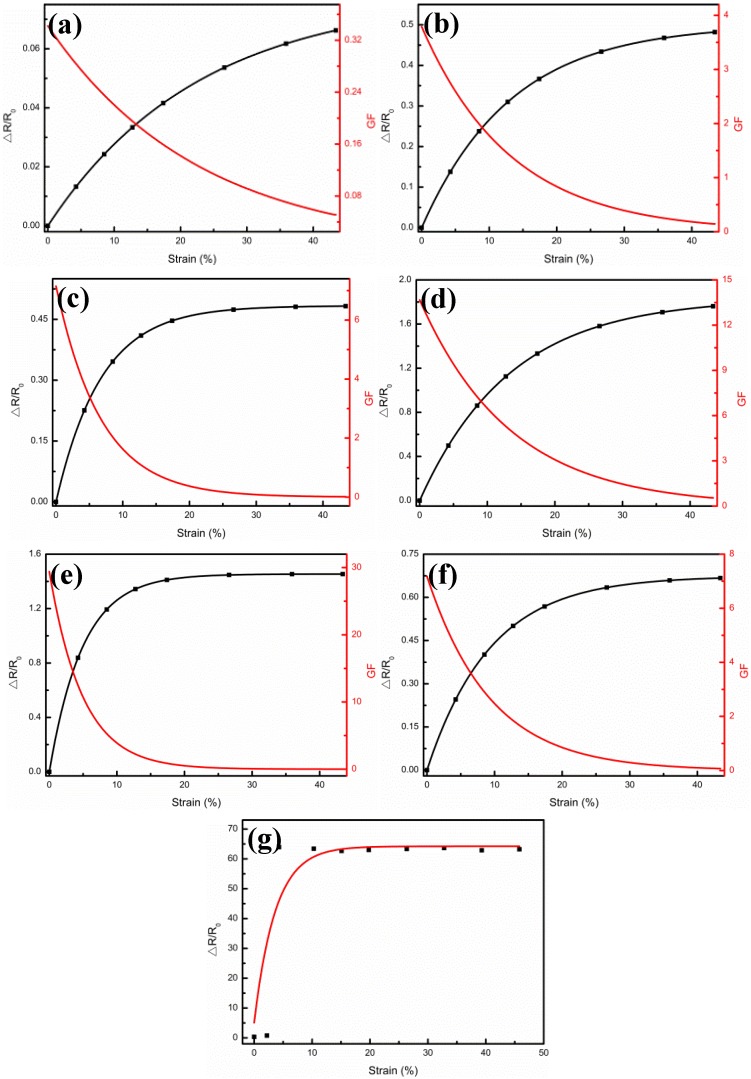
Relative resistance (ΔR/R_0_) variations and gauge factors (GF) versus strains from 0% to 50% at different Ag film thicknesses. (**a**) 0.5 μm; (**b**) 1.5 μm; (**c**) 1.75 μm; (**d**) 3.26 μm; (**e**) 4.9 μm; (**f**) 6.4 μm; (**g**) ΔR/R_0_ of the pure Ag film as a function of strain from 0% to 50%.

**Figure 7 nanomaterials-09-00850-f007:**
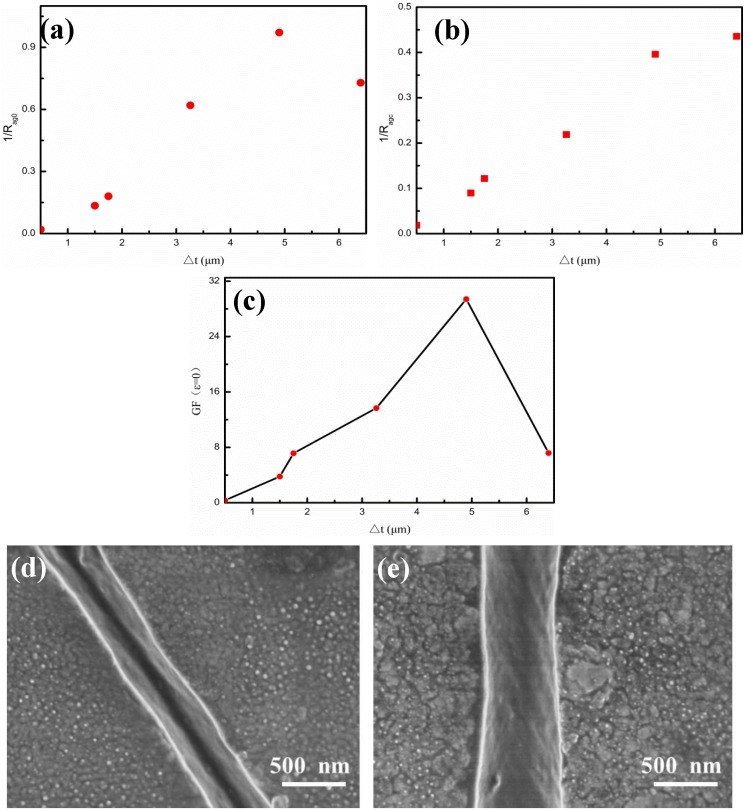
Effects of the Ag film thicknesses (Δt) on the sensitivity of the Ag/CNTs-PDMS composite film sensor. (**a**) 1⁄R_Ag0_-Δt; (**b**) 1⁄R_Agc_-Δt; (**c**) Gauge factors (GF)-Δt; (**d**) the Ag film cracked under conditions of 4.9 μm thickness and 5% strain; (**e**) the Ag film cracked under conditions of 6.4 μm thickness and 5% strain.

**Figure 8 nanomaterials-09-00850-f008:**
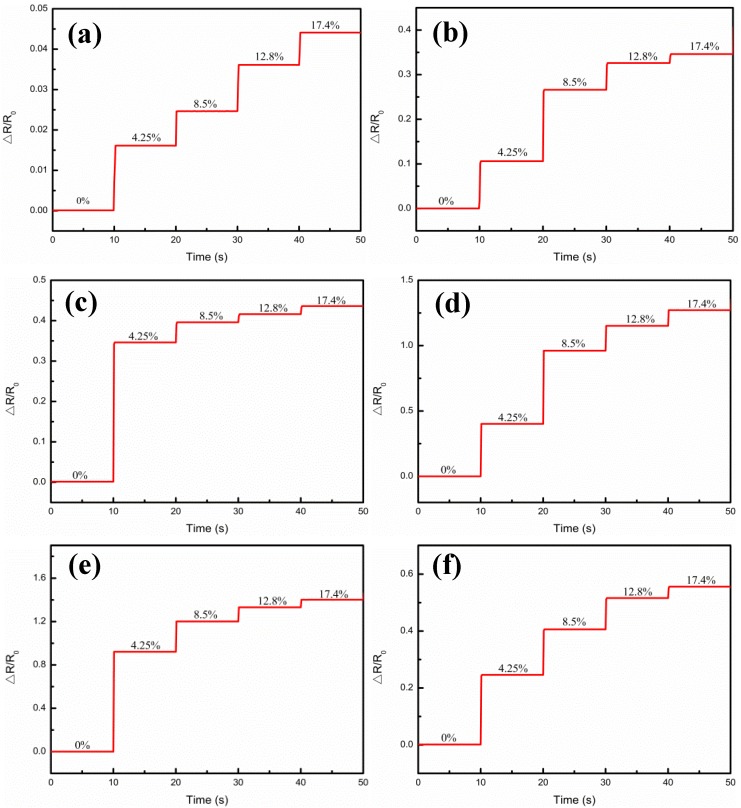
Relative resistance (ΔR/R_0_) variations under step strains from 0% to 17.4% at different Ag film thicknesses. (**a**) 0.5 μm; (**b**) 1.5 μm; (**c**) 1.75 μm; (**d**) 3.26 μm; (**e**) 4.9 μm; (**f**) 6.4 μm.

**Figure 9 nanomaterials-09-00850-f009:**
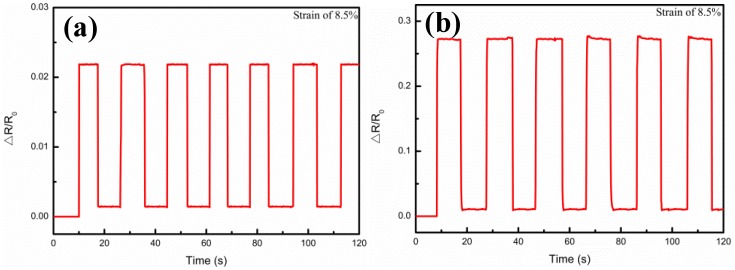
Multicycle tests of relative resistance (ΔR/R_0_) variations under periodic strain up to 8.5% at different Ag film thicknesses. (**a**) 0.5 μm; (**b**) 1.5 μm; (**c**) 1.75 μm; (**d**) 3.26 μm; (**e**) 4.9 μm; (**f**) 6.4 μm.

**Figure 10 nanomaterials-09-00850-f010:**
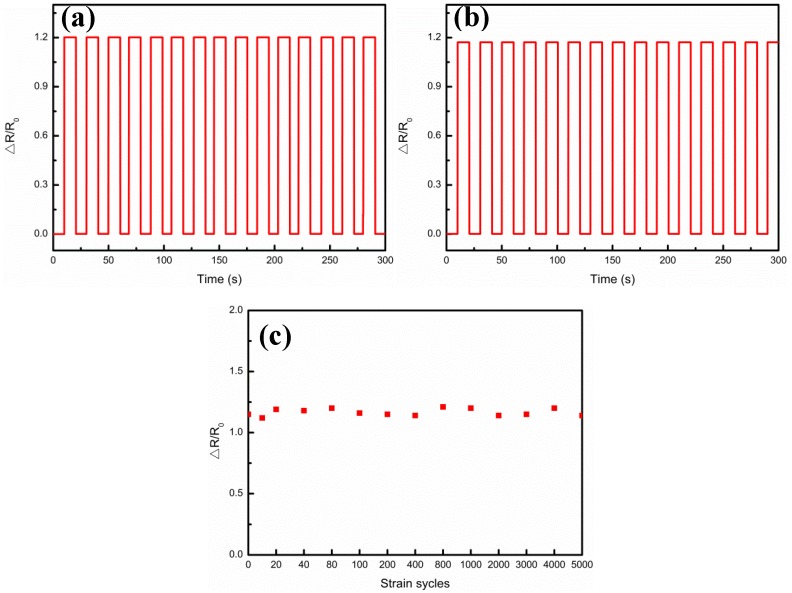
Relative resistance (ΔR/R_0_) variations under conditions of Al film thickness 4.9 μm and strain 8.5% at 0.05 Hz frequency as a function of the strain cycle. (**a**) ΔR/R_0_ at the beginning; (**b**) ΔR/R_0_ at the end; (**c**) ΔR/R_0_ after 5000 cycles.

**Figure 11 nanomaterials-09-00850-f011:**
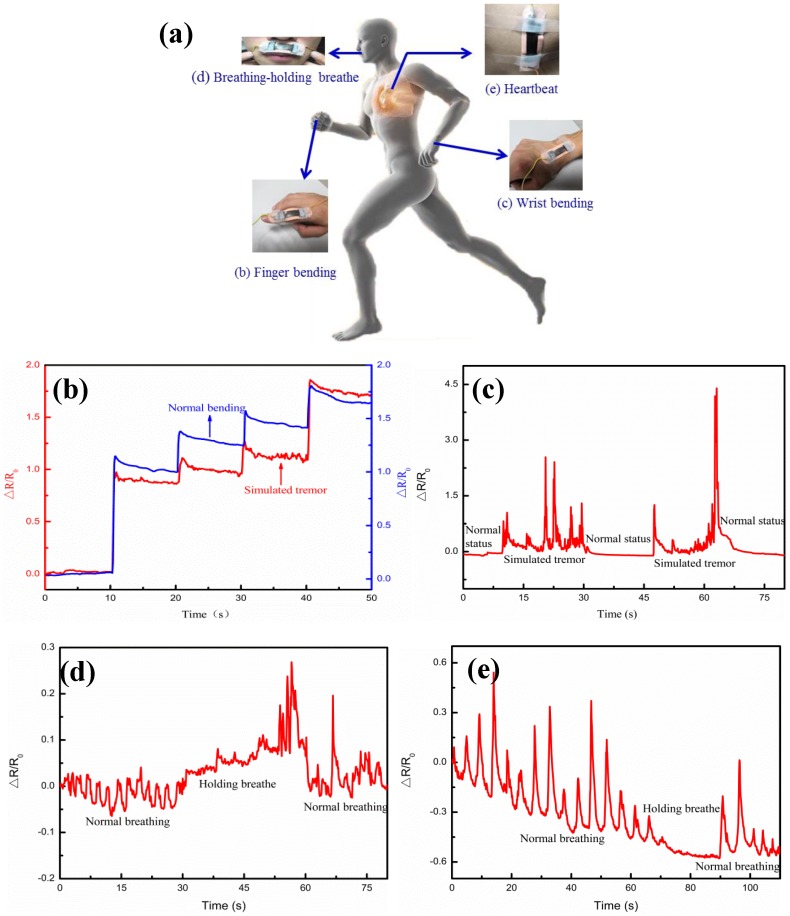
Application of the flexible Ag/CNTs-PDMS composite film sensor for monitoring weak signals from the human body. (**a**) Overview of the sensor’s locations; (**b**) signals from finger-bending, involving normal bending (blue line) and simulated tremor (red line); (**c**) signals from the wrist under conditions of normal state and simulated tremor; (**d**) signals from the upper lip under conditions of normal breathing and holding breath; (**e**) signals from the chest under conditions of normal breathing and holding breath.

**Table 1 nanomaterials-09-00850-t001:** Relationships between the resistance and parameters including length, width, and thickness at different sputtering times.

Time (min)	LengthL (cm)	Widthw (cm)	Thickness∆t (μm)	ResistanceR_0_ (kΩ)
0.5	1	0.5	0.5	3.6
1	1.5	2.5
2	1.75	1.75
4	3.26	1.05
6	4.9	0.60
8	6.4	0.55
CNTs	0.732		
Pure Ag	1.12

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
