# Peer review of "A Wrinkled Ag/CNTs-PDMS Composite Film for a High-Performance Flexible Sensor and Its Applications in Human-Body Single Monitoring"

_nanomaterials, 2019, doi:10.3390/nano9060850_

Reviewer 1 Report

The manuscript describes a study of a mechanosensor consisting of a wrinkled carbon nanotube film on PDMS sustrate with sputtered deposited silver overlayer. The silver film thickness is the main parameter investigated.

There is uncertainty about terminology throughout the manuscript. It begins in the abstract and on line 75 with terms such as "force electric effect", "wrinkled structure", "wave structure" and "force-electric conversion effect". All the quotation marks should be removed. If there is uncertainty in what the terms describe, then they should be replaced. Wrinkled should be used to describe the structure since wave implies uniformity which is not seen in Figure 2a.

The original D Kang et al. (2014) spider mimicry publication is cited in reference 21 but should be included in the reference list in the manuscript. 

The authors motivate the study as an extension of their previous work, which used gold as transducer as described in reference 21, based on silver having a lower price and thus being more suitable for large scale production. Of more importance at the development stage is how silver sensor performance compares with other similar devices. Numbers are required, not just the generalities given on line 102. Some discussion of if gauge factor, given as 0.8 in reference 25, 70 in reference 21 and 2000 in the Kang article, can be used to compare sensors should be added to the manuscript. If silver/gold material cost is important, then some mention should be made of the nanometer thicknesses of gold used in sensors described in E Lee et al.,Effect of Metal Thickness on the Sensitivity of Crack-Based Sensors, Sensors 18 (2018) 2872. That silver has a lower price does not compensate for the much thicker films described in the manuscript.  Maximal sensor strain is compared with metal sensors on line 233 but not with flexible sensors, for example, 280% in reference 25.

Figure 10d seems missplaced since it doesn't deal with long term stability. It is not only a carbon nanotube film which can provide mechanical stability for a transducer layer. Direct adhesion of silver to PDMS can be improved by pretreatment of the polymer as described in J Cui et al., Flexible Pressure Sensor with Ag Wrinkled Electrodes Based on PDMS Substrate, Sensors 16 (2016) 2131.

It is not clear if it is gas flow causing the sensor response in Figure 11d as stated on 290. Temperature sensitivity of the sensor has been described in reference 21 and moisture in exhaled breath can also affect resistance.

The manuscript has value in its current form but can be improved by a more extensive comparision of the results with the authors' previous work and other similar sensors in the literature.

Author Response

Dear Reviewer,

Thanks for your comments on our manuscript. We have modified the manuscript accordingly. please check the PDF.

Reviewer 2 Report

In this paper, the authors reported “A Wrinkled Ag/CNTs-PDMS Composite Film for High-performance Flexible Sensor and Its Applications in Human-body Single Monitoring”. The work suggested impacts on fabrication of a flexible sensor and its application about healthcare. In my opinion, this manuscript is maybe useful for this journal and near fields. However, in main manuscript, there are not detail information about your work. This is not faithfulness for paper readers. Also, there are unclear points as following;

1.           In main manuscript, there are no description about Figures. For example, in Figure 10, the authors mentioned detailed explanation about only Figure 10 (d). Thus, I cannot understand detailed your work. In other figures, Figure 4, 6, 8, and 9 are same.

2.           In Figure 4 (b) and (d) are indistinct and unclear. The authors should change these to better ones. Also, there is no description about each layers. Where is Ag? Where is CNTs-PDMS?

3.           In Figure 11, the authors tried to monitor signals of human. Especially, Figure 11 (e) showed signals of human breathing. In here, why baseline shifted from 0.0 to -0.5 in a R/R0 axis?

Author Response

Dear Reviewer,

Thanks for your comments on our manuscript. We have modified the manuscript accordingly. please check the PDF.

Round  2

Reviewer 1 Report

One of the new text sections which begins on line 237 could be corrected as the following:

Compared to other sensors, the present Ag/CNTs-PDMS composite sensor achieved sensitivity up to 30 and withstood strain up to 50%, which makes it more suitable for application as a wearable strain sensor in terms of cost, sensitivity, and preparation process. For example, Takeo Yamada et al [27] prepared a stretchable carbon nanotube strain sensor for human-motion detection, which exhibited advantages such as strain toleration up to 280%, high durability, fast response and low creep, but the sensitivity (GF=0.8) was low. Daeshik Kang et al [3] proposed an ultrasensitive mechanical crack-based sensor inspired by the spider sensory system with sensitivity up to 2000, but it tolerated strain only up to 2%. In our previous work about a Au/CNTs-PDMS composite film sensor [22], sensitivity was up to 70, but the Au transducer layer employed was more expensive than the present Ag one.

Author Response

We have corrected our manuscript.

Reviewer 2 Report

The problems that I mentioned have been revised.

I agree to be publish this manuscript.

Author Response

Thanks for your comments!
